# Positive Association between Preserved C-Peptide and Cognitive Function in Pregnant Women with Type-1 Diabetes

**DOI:** 10.3390/biomedicines10112785

**Published:** 2022-11-02

**Authors:** Marina Ivanisevic, Darko Marcinko, Sandra Vuckovic-Rebrina, Josip Delmis

**Affiliations:** 1Department of Obstetrics and Gynecology, School of Medicine, University Hospital Centre Zagreb, University of Zagreb, 10000 Zagreb, Croatia; 2Clinic for Psychiatry and Psychological Medicine, School of Medicine, University Hospital Centre Zagreb, University of Zagreb, 10000 Zagreb, Croatia; 3Vuk Vrhovac Clinic for Diabetes, Endocrinology and Metabolic Diseases, Merkur University Hospital, 10000 Zagreb, Croatia

**Keywords:** cognitive function, C-peptide, pregnancy, type 1 diabetes mellitus

## Abstract

This study focused on the cognitive function of women with type 1 diabetes in pregnancy. We investigated risk factors for a low cognitive score such as age, duration of Diabetes, BMI, subclinical hypothyroidism, cardiovascular autonomic neuropathy, the impact of hypo-/hyperglycemia, and C-peptide preservation. Material and methods. Seventy-eight pregnant women with type 1 diabetes (age 31.1 ± 5.4 years, diabetes duration 14.3 ± 8.9 years) were included in the study. Cognitive function was assessed in different domains, such as reasoning, memory, attention, coordination, and perception. Results. The cognitive test values ≥400 were considered high scores, and values <400 were considered low. Relative risks for low scores for general cognitive function were associated with increased BMI > 25 kg/m^2^ 2.208 (95% CI 1.116–4.370), HbA1c > 6.5% RR 0.774 (95% CI 0.366–1.638), subclinical hypothyroidism RR 3.111 (95% CI 1.140–8.491), and impaired cardiovascular autonomic neuropathy RR 2.250 (95% CI 1.000–5.062). Pregnant women with a lower score for general cognitive function had higher BMI and higher leptin levels. Preserved C-peptide reduces the risk for cognitive impairment (RR 0.297 (95% CI 0.097–0.912)) in pregnant women with type 1 diabetes Conclusion. BMI > 25 kg/m^2^, subclinical hypothyroidism, and cardiovascular autonomic neuropathy are associated with increased risk, and postprandial C-peptide preservation with reduced risk for cognitive impairment in pregnant women with type 1 diabetes.

## 1. Introduction

Type 1 diabetes (T1DM) is a consequence of the immune-mediated destruction of β-cells with consequent loss of the ability to produce and secrete insulin [1]. T1DM in pregnant people has numerous risks for the mother and the newborn [2,3]. Cognitive functions are mental processes that enable a person to play an active role in perception, learning, memory, attention, and information processing. Previous research has shown that patients with type 1 diabetes usually have reduced performance in learning and memory, attention, information processing speed, and visual perception compared to the non-diabetic population [4]. The severity of cognitive impairment in patients with type 1 diabetes is influenced by the patient’s age and duration of diabetes [4,5,6].

The authors of this study found no published research on cognitive function in pregnant women with type 1 diabetes. Research on cognitive function mainly focuses on children and adults born to mothers with T1DM [7,8,9,10,11]. Children born to mothers with T1DM have an increased risk of developing long-term metabolic and neurocognitive disorders compared with children born to non-diabetic mothers. The pathogenesis of cognitive impairment in people with T1DM is associated with changes in glucose metabolism in the brain and damage to blood vessels (microangiopathy).

Research has shown that hypoglycemia and hyperglycemia are causes of cognitive dysfunction [12,13]. Cameron FJ et al. [14] believe that chronic hyperglycemia is more harmful to the development of cognitive dysfunction than hypoglycemia.

Patients with diabetes complications, such as proliferative retinopathy, nephropathy, neuropathy, hypertension, and cardiovascular autonomic neuropathy, have poorer cognitive function compared to those without complications [15]. The development of the aforementioned complications and changes in the brain depends on the duration of diabetes and the quality of metabolic control. Based on previous research, it can be concluded that cognitive impairment is one of the complications of type 1 diabetes.

Sima A.A. et al. [16] found apoptotic loss of neurons in rats with type 1 diabetes due to impaired insulin action. C-peptide is a bridging peptide of proinsulin and a link between the A and B chains of insulin. Insulin/C-peptide deficiency in type 1 diabetes is an essential factor in the pathogenesis of CNS dysfunction [17]. C-peptide substitution partially prevents hippocampal neuronal apoptosis and cognitive deficits [16]. The authors believe that C-peptide has a significant function in supporting insulin action with multiple beneficial effects on diabetic polyneuropathy and diabetic encephalopathy in type 1 diabetes. Thyroid hormones are crucial for normal brain function [18,19]. Cardiovascular autonomic neuropathy is the impairment of autonomic control of the cardiovascular system [20]. It is often diagnosed in people with type 1 diabetes, especially those with a long disease duration. Cardiovascular autonomic failure is associated with the onset of cognitive dysfunction [21]. Poor nutrition and obesity are associated with cognitive impairment in adulthood and an increased risk of dementia [22]. Increasing body weight decreases insulin sensitivity, leading to a metabolic disorder [23]. Leptin is essential in regulating food intake and energy balance in adults and acts as a satiety factor in the adult brain [24].

This study aimed to investigate the impact of age, diabetes duration, BMI, subclinical hypothyroidism, cardiovascular autonomic neuropathy, hypo-/hyperglycemia, and C-peptide preservation on cognitive performance.

## 2. Materials and Methods

### 2.1. Ethical Statements

The Ethics Committee of the School of Medicine, the University of Zagreb, approved the study (No. 380-59-10106-19-111/26) within the scientific project PRE-HYPO No. IP-2018-01-1284. All women in the study provided informed consent for themselves and their newborns.

### 2.2. Study Participants

#### 2.2.1. Inclusion Criteria

In the prospective observational cohort study, we consecutively included 78 women with type 1 diabetes mellitus before completing ten gestational weeks during the study period from 1 February 2019, to 31 December 2021. At ten weeks of gestation, women underwent a cognitive function test.

Cognitive function was assessed in different domains, such as reasoning, memory, attention, coordination, and perception (https://www.cognifit.com, accessed on 31 December 2021). A score between 0–200 is considered as cognitive weakness, and a score between 200–400 is low, although within the average. A score between 400–600 is a high score, and a score between 600–800 is above the norm. The cognitive abilities with this score are considered strengths. Since a small number of participants had a score of 0–200 or a score of 600–800, we divided the Cognifit test values (scores) into two groups, a high score group with a score ≥400 and a low score group whose results were <400, Appendix A.

#### 2.2.2. Exclusion Criteria

Exclusion criteria for the PRE-HYPO study participants were: proliferative retinopathy, nephropathy, and chronic hypertension.

### 2.3. Data Collection

#### 2.3.1. Cognitive Test

Cognifit (Cognitive Assessment battery) was used to test cognitive function. The test consists of the following areas: reasoning, memory, attention, coordination, and perception. This program evaluates a wide range of abilities and assesses cognitive well-being (high-moderate-low), identifying strengths and weaknesses in memory/attention, executive functions, planning, and coordination. Reasoning consists of the following subdomains: planning, processing speed, and shifting. The subdomains for memory are short-term phonological memory, contextual memory, naming, short-term memory, non-verbal memory, visual short-term memory, and working memory. Attention consists of divided attention, focused attention, inhibition, and updating. Coordination encompasses visual-motor coordination (hand-eye coordination) and reaction speed (response time). Perception includes auditory perception, estimation, recognition, spatial perception, visual perception, and visual scanning. We defined general cognitive functioning as comprising of the following areas: reasoning, memory, attention, coordination, and perception.

#### 2.3.2. Continuous Glucose Monitoring

Continuous glucose monitoring (CGM) monitors glucose levels in the interstitial fluid to improve metabolic control. CGM measurement was done on the iPro2 device (professional CGM system “Blinded”). The components of the iPro2 system are the sensor and the transmitter. The iPro2 sensor provides a retrospective picture of glycemia for seven days with no alarm or monitor. Interstitial glucose levels are continuously measured, but as data is not continuously transmitted from the sensor, results are available after scanning the sensor with a reading device.

#### 2.3.3. Subclinical Hypothyroidism in Pregnant Women with Type 1 Diabetes

Subclinical hypothyroidism is defined based on the TSH value. If the TSH value was less than 2.5 IU/L, we considered it euthyroidism, and for values ≥2.5 IU/L in the first trimester of pregnancy, and values of TSH ≥ 3.0 IU/L in the second and third trimester, we considered it to be subclinical hypothyroidism.

#### 2.3.4. Cardiovascular Autonomic Neuropathy

Cardiovascular autonomic neuropathy in the first trimester of pregnancy was diagnosed with the computer system Vagus 2100. Testing was performed using a computer system based on heart rate variability (R-R interval in the electrocardiogram) during rest, deep breathing, Valsalva test, active orthostatic test (Ewing’s test), and orthostatic blood pressure response. After statistical processing of data by standard, vector, and spectral analysis, the following parameters were obtained: coefficient of variation of R-R-interval at rest, the spectrum of very low frequency, low frequency, and high frequency of variation in spectral analysis, coefficient of variation of R-R-interval during deep breathing, E/I-ratio, 30: 15 ratio, Valsalva ratio, and orthostatic response. According to the test results, CAN was divided into two groups, a group without CAN damage and a group with CAN damage.

#### 2.3.5. Blood Sample Analyses

The maternal vein sera were analyzed for fasting leptin and postprandial C-peptide concentration, and HbA1c percentage was measured in maternal blood only.

The hexokinase method quantified glucose levels on a Cobas C301 analyzer with reagents from the same manufacturer (Roche, Basel, Switzerland). The HbA1c levels in whole blood were measured by turbidimetric inhibition immunoassays on a Cobas C501 instrument (Roche, Basel, Switzerland). The C-peptide concentrations were determined by electrochemiluminescence immunoassays (ECLIA) with Elecsys immunoassay analyzers (Roche Diagnostics, Switzerland). The lower detection limit of C-peptide in serum is 0.003 nmol/L.

The leptin serum concentration was determined by sandwich Kit, Tecan (Männedorf, Switzerland), IBL. International (Hamburg, Germany) (Cat.No.MD53001).

The following parameters were recorded: maternal height (cm) and weight (kg) before pregnancy, and prepregnancy body mass index (kg/m^2^; BMI), calculated from the prepregnancy values.

#### 2.3.6. Sample Size

For sample size, we used a correlation between C-peptide and total cognitive function score in the first trimester of pregnancy. For a sample size of 27, α level of 0.05, a power (1-β) was 0.9518067.

#### 2.3.7. Statistical Analysis

Statistical analyses were performed using the statistical package SPSS version 24 (IBM, Armonk, NY, USA). Absolute and relative frequencies represent categorical data. The Kolmogorov–Smirnov test tested normality of distribution. Numerical data are described by the mean and standard deviation in the distributions following the normal, and in other cases, by the median and the interquartile range’s limits. Student’s *t*-test tested group differences between normally distributed continuous variables, and differences between nonnormally distributed continuous variables were tested by the Mann-Whitney U test. The χ2 test and risk ratio (RR) trial tested group differences between categorical variables. Pearson’s correlation coefficient (r) evaluated the correlation between normally distributed numerical variables. For relative risk (RR), and heterogeneity we used the statistical package Comprehensive meta-analyses version 3 (Biostat, Englewood, NJ, USA). All *p* values are two-sided. The significance level was set at *p* < 0.05.

## 3. Results

### 3.1. General Data

Table 1 shows the demographic data of pregnant women. The majority of pregnant women (61.5%) are over 30 years old, with the onset of the disease after ten years of age (66.7%), with a longer duration of the disease than eight years (70.5%). There were 25 overweight and obese pregnant women (32.1%). A total of 53.8% of them had a high school diploma. Furthermore, 53.8% had HA1c > 6.5%, 51.4% had subclinical hypothyroidism, and 43.6% had impaired cardiovascular autonomic neuropathy. Of the total group, 13 (16.7%) pregnant women had detectable C-peptide in the first trimester of pregnancy. Finally, 28.1% of pregnant women had CGM values of more than 7.8 mmol/L, and 71.9% had a value of less than 3.9.

### 3.2. Results of Cognitive Function Test According to High—Moderate—Low Scores

Table 2 shows the results of the cognitive function test. In the first trimester of pregnancy, scores above the average (600–800) were achieved for: reasoning 5 (6.4%), memory 2 (2.6%), attention 28 (35.8%), coordination 1 (1.3%), perception 7 (9.0%), and the total cognitive function of 3 (3.8%) pregnant women. The largest number of pregnant women had a high score (400–600) for all areas of cognitive function. A total of 25 (32.1%) had a low score below the average for the area of coordination and 2 (2.6%) for the area of memory. Finally, 39 (50.2%) pregnant women had a low score (200–400) within the average for the area of memory and 35 (44.9%) for the area of coordination.

### 3.3. Mean and Standard Deviation of Cognitive Tests

Table 3 shows the lowest, highest, and mean value (score) with a standard deviation of individual areas and the total cognitive test in the first trimester of pregnancy. Pregnant women with T1DM had the best results in attention and reasoning and the worst in coordination and perception. The mean value for the overall effect (score) of cognitive function was 440.7 ± 83.2 for the first trimester.

### 3.4. Relative Risks for Cognitive Functions

BMI > 25 kg/m^2^ increases the risk of memory, coordination, and total cognitive function decline. Hypothyroidism increases the risk of overall cognitive function impairment. Preserved C-peptide reduces the risk of potential decline in reasoning, memory, coordination, and general cognitive function. Cardiovascular autonomic neuropathy (CAN) increases the risk of impaired reasoning, memory, attention, coordination, and total cognitive function.

In the first trimester of pregnancy, BMI > 25 kg/m^2^, subclinical hypothyroidism, and cardiovascular autonomic neuropathy increase the risks, and postprandial C-peptide preservation reduces the chances of a low score for total cognitive function and memory in pregnant women with T1DM (Table 4 and Figure 1 and Figure 2).

Mothers with a low cognitive function score had a significantly higher concentration of leptin in their blood and BMI. (Figure 3 and Appendix A).

## 4. Discussion

To our knowledge, this is the first study to assess the cognitive function of pregnant women with type 1 diabetes. Cognitive impairment is considered a complication of type 1 diabetes. Patients with type 1 diabetes typically have reduced performance in the following cognitive domains: learning and memory, attention, processing speed information, and visual perception concerning the non-diabetic population [6]. The severity of cognitive impairment in patients with type 1 diabetes is influenced by the patient’s age, onset, and duration of diabetes. In the first trimester of pregnancy, no increased risk was found for low scores for cognitive functions concerning age, onset, and duration of diabetes. Cognitive decline in T1DM is more expressed in adults with T1DM than in children with T1DM [4,25]. In patients with a longer duration of diabetes, cognitive function impairment is more pronounced. Patients with early-onset diabetes are likelier to have lower cognitive test scores.

The pathogenesis of cognitive impairment in people with T1DM is associated with changes in glucose metabolism in the brain and damage to blood vessels (microangiopathy) [11]. Research has shown that both hypoglycemia and hyperglycemia are involved as causes of cognitive dysfunction.

HbA1c is an essential biomarker in pregnant women with T1DM to evaluate glycemic status for the last three months. The pregnant women in this study had reasonable glycemic control; the average value of HbA1c in the first trimester was 6.9 ± 1.3 mmol/L. During the first trimester of pregnancy, glycemia was monitored by CGM for 14 days. Although a high percentage of pregnant women had a glucose concentration ≤3.9 mmol/L, none had severe hypoglycemia. A lower rate of pregnant women had CGM values >7.8 mmol/L.

Most often, T1DM is also diagnosed with thyroid disease, i.e., altered values of thyroid hormones. Common autoimmune and non-autoimmune causes can cause hypofunction or hyperfunction of one or more endocrine glands. A total of 12 (15.4%) pregnant women had Hashimoto’s thyroiditis, and four (5.1%) pregnant women had hyperthyroidism. In the first trimester of pregnancy, subclinical hypothyroidism was diagnosed in 51.4% of pregnant women. Pregnant women with subclinical hypothyroidism had one spontaneous abortion at 12 weeks gestation and three premature deliveries. Pregnant women without subclinical hypothyroidism had five premature births. A significant difference was found in pregnancy outcomes between women with subclinical hypothyroidism and those without it. According to the results of other authors, pregnant women with subclinical hypothyroidism during pregnancy are more prone to glucose and lipid metabolism disorders, which increases the risk of miscarriage, premature birth, preeclampsia, and PROM at term. Subclinical hypothyroidism during pregnancy is a risk factor for a low score of overall cognitive function and after childbirth for memory, attention, and general cognitive function. Our research confirmed the influence of subclinical hypothyroidism on the cognitive function of pregnant women, in agreement with the investigation of Pop VJ et al. [19]. Thus it is crucial to determine pregnancy TSH levels, especially in women with T1DM. In women not diagnosed with thyroid dysfunction before pregnancy, TSH and FT4 and thyroid antibodies should be determined at pregnancy confirmation. [26,27]. To improve maternal and neonatal outcomes, women with subclinical hypothyroidism need levothyroxine replacement therapy. Any form of hypothyroidism, if left untreated, can impair glycemic control and lipid metabolism in pregnant women with type 1 diabetes.

Cognitive function is significantly worse in pregnant women than in the control group women, which some authors associate with an increase in hormones, progesterone, estrogen, human placental lactogen (HPL), cortisol, prolactin (PRL), and others [28]. Our participants with T1DM had their cognitive function tested in the first trimester of pregnancy due to the more frequent occurrence of hypoglycemia than in the second and third trimesters. However, the study did not show an association between lower glucose values measured by CGM and a higher frequency of lower scores for cognitive function.

Obesity is a significant risk factor for cognitive functioning. Of the 78 pregnant women with T1DM, 12 (15.4%) were obese (BMI ≥ 40 kg/m^2^). Obese pregnant women with the onset of T1DM before age 10 had significantly lower values of cognitive function. Overweight and obese pregnant women are at significant risk for decreased overall cognitive function, memory, and coordination. Mothers with low overall cognitive function and memory values had a significantly higher concentration of leptin in their blood. Other authors presented similar results in middle-aged individuals [29]. Comparing leptin concentration and total cognitive function score, a negative correlation was found (*r* = −0.349, *p* = 0.001; Appendix A). A lower leptin level with a higher cognitive function score is shown in research by Warren MW et al. [30]. Leptin is an adipostatic hormone that is released into circulation depending on the amount of adipose tissue [24,31]. It participates in the regulation of body weight through the suppression of appetite and stimulation of energy consumption [24]. Leptin is strongly correlated with maternal weight and BMI and serves as a biomarker of maternal and fetal obesity.

Pregnant women with detectable C-peptide had a significantly reduced risk for diminished reasoning, memory, coordination, and full cognitive function. C-peptide is a peptide consisting of 31 amino acids that, in the proinsulin molecule, connect the amino end of the alpha chain and the carboxyl end of the beta chain of insulin, hence the name connecting peptide (C-peptide) [32]. Insulin and C-peptide are excreted together in equimolar amounts into the portal circulation. Residual insulin plays an essential regulatory role in peripheral tissues and in CNS [32]. Primary diabetic encephalopathy is caused by hyperglycemia and impaired insulin action, which develops depending on the duration of diabetes and is associated with apoptosis, loss of neurons, and decline in cognitive function [33]. The aforementioned appears to be particularly associated with type 1 diabetes. This finding is consistent with data showing that the antiapoptotic effects of insulin and C-peptide are mediated via PI-3 kinase stimulation, p38 activation, disinhibition of inhibitor of B, translocation of NF-B, promotion of Bcl2, and inhibition of jun NH2-terminal kinase phosphorylation [34,35]. C-peptide replacement prevented the decrease in endogenous NGF and NGFRtryosine receptor kinase [34,35]. Impaired C-peptide activity in insulin-dependent type 1 diabetes plays a prominent role in primary diabetic encephalopathy. C-peptide in type 1 diabetes has a protective effect on the cognitive deficit.

The limitation of the study is the absence of a control group, that is, healthy pregnant women. There is no comparison of cognitive function between pregnant women before and during pregnancy or postpartum. According to the available literature, there is no data on the cognitive function of pregnant women with type 1 diabetes. For this reason, the authors did not have the opportunity to compare this study’s results with other research results. For memory and overall cognitive function, heterogeneity was I^2^ 48.6% and 40.7%, which represents moderate heterogeneity and might be considered a limitation of the meta-analysis.

## 5. Conclusions

BMI ≥ 25 kg/m^2^, subclinical hypothyroidism, and cardiovascular autonomic neuropathy are associated with increased risk, and C-peptide preservation reduces the risk for cognitive impairment in pregnant women with type 1 diabetes. Good metabolic control, normal BMI and thyroid function, and preserved C-peptide are important for proper cognitive function in pregnant women with T1DM.

## Figures and Tables

**Figure 1 biomedicines-10-02785-f001:**
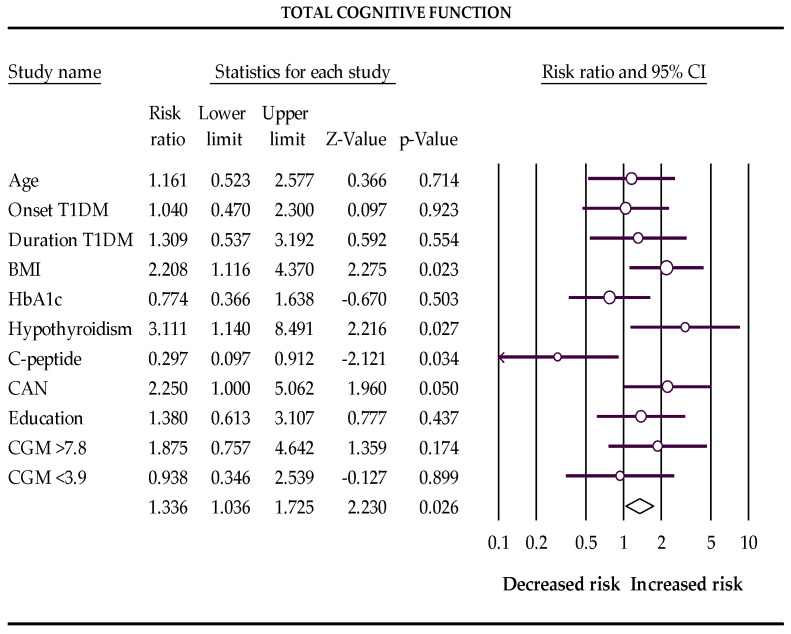
The Forest plot represents the relative risks for total cognitive function (Q-value 16.862, df 10, *p*-value 0.077, I^2^ 40.695). The following data are presented: age, age of onset disease, duration of disease, BMI, HbA1c %, subclinical hypothyroidism, C-peptide preservation, cardiovascular autonomic neuropathy (CAN), level of education, percentage of glucose concentration >7.8 mmol/L, and percentage of glucose concentration <3.9 mmol/L measured with a continuous glucose monitor (CGM). Empty circles represent the relative risk of effect for each category; horizontal lines around circles indicate confidence intervals, and the central vertical line represents the magnitude of the null effect.

**Figure 2 biomedicines-10-02785-f002:**
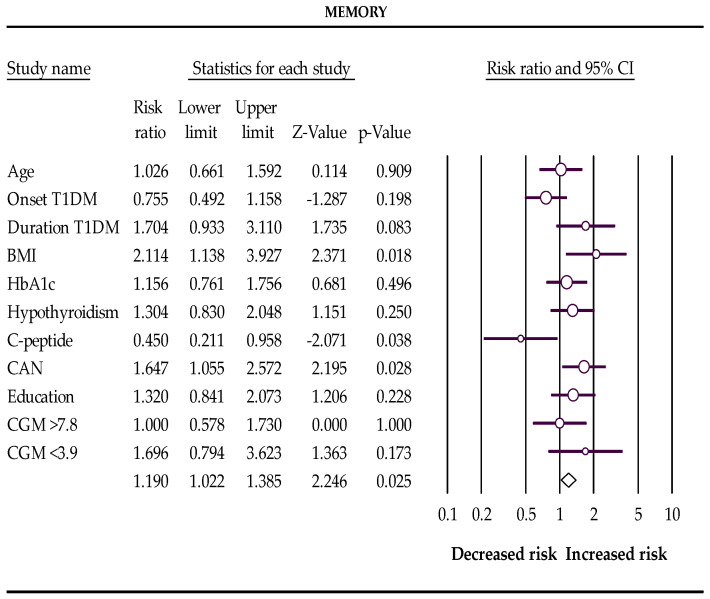
The Forest plot represents the relative risks for memory (Q-value 19.464, df 10, *p*-value 0.035, I^2^ 48.623). The following data are presented: age, age of onset disease, duration of disease, BMI, HbA1c %, subclinical hypothyroidism, C-peptide preservation, cardiovascular autonomic neuropathy (CAN), level of education, percentage of glucose concentration >7.8 mmol/L, and percentage of glucose concentration <3.9 mmol/L measured with a continuous glucose monitor (CGM). Empty circles represent the relative risk of effect for each category; horizontal lines around circles indicate confidence intervals, and the central vertical line represents the magnitude of the null effect.

**Figure 3 biomedicines-10-02785-f003:**
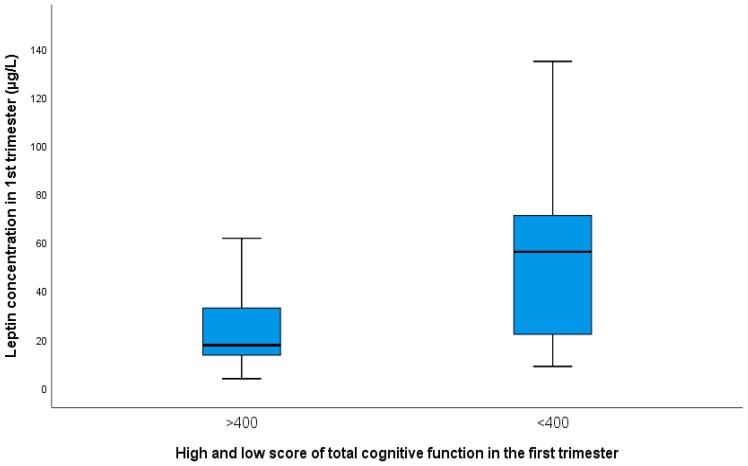
Presentation of mean values with SD of leptin in groups of pregnant women with a higher score for total cognitive function (24.0 ± 6.20) and with a low score (52.60 ± 41.50; *p* = 0.013).

**Table 1 biomedicines-10-02785-t001:** Patients’ demographic data in the first trimester.

Variable	N (%)	Minimum	Maximum	Mean ± SD or Median and IQR
Age (years)	78	19	38	30.7 ± 5.5
<30 years	30 (38%)	19	29	25.1 ± 3.2
≥30 years	48 (61.5%)	30	38	34.4 ± 2.9
Duration of T1DM	78	2.0	36	14.5 ± 8.6
<8 years	23 (29.5)	2	7	3.9 ± 2.2
≥8 years	55 (70.5)	9	36	18.8 ± 6.2
Age of onset T1DM	78	2	32	16.3 ± 9.2
Before 10 years	26 (33.3)	2	9	7.4 ± 2.7
After 10 years	52 (66.7)	11	32	21.4 ±7.5
Height (cm)	78	156	183	166.1 ± 5.7
BMI (kg/m^2^) before pregnancy	78	15.1	40.3	24.3 ± 5.3
≤24.9 kg/m^2^	53 (67.9)	15.1	24.9	21.2 ± 2.0
>25 kg/m^2^	25 (32.1)	25.6	40.3	30.8 ± 4.4
Education	78			
High and university degree	36 (46.2)			
Graduate school degree	42 (53.8)			
HbA1c (%)	78	5.4	10.1	6.9 ± 1.3
≤6.5%	36 (46.2)	5.4	6.4	6.0 ± 0.3
≥6.5%	42 (53.8)	6.5	10.1	7.7 ± 1.4
Hypothyroidism	74	0.5	5.1	2.4 ± 1.2
No (TSH <2.5)	36 (48.6)	0.5	2.4	1.6 ± 0.7
Yes (TSH >2.5)	38 (51.4)	2.5	5.1	3.5 ± 0.9
CAN	78			
Normal	44 (56.4)			
Impaired	34 (43.6)			
C-peptide	55			
Negative (<0.03 nmo/L)	13 (23.6)			
Positive (≥0.03 nmol/L)	42 (76.4)	0.03	0.9	0.18 (0.1–0.28)
CGM mean glucose	65	4.0	11.8	6.2 ± 1.3
CGM ≥ 7.8 mmol/L n (%)	65			
No	46 (71.9)			
Yes	19 (28.1)			
CGM target range				
3.9–7.8 mmol/L n (%)	65			
No	50 (76.6)			
Yes	15 (23.4)			
CGM ≤ 3.9 mmol/L n (%)	65			
No	19 (28.1)			
Yes	46 (71.9)			

CAN, cardiovascular autonomic neuropathy; CGM, continuous glucose monitoring; TSH, thyroid stimulating hormone.

**Table 2 biomedicines-10-02785-t002:** Results of the cognitive function test (Cognifit) in the first trimester.

The First Trimester n = 78				
Domain/Score	600–800	400–600	200–400	0–200
Reasoning	5 (6,4%)	56 (71.8%)	17 (21.8%)	
Memory	2 (2.6%)	35 (44.9%)	39 (50.2%)	2 (2.6%)
Attention	28 (35.8%)	43 (55.1%)	7 (9.0%)	
Coordination	1 (1.3%)	17 (21.8%)	35 (44.9%)	25 (32.1%)
Perception	7 (9.0%)	53 (67.8%)	18 (23.1%)	
Total cognitive function	3(3.8%)	55 (70.6%)	20 (25.6%)	

**Table 3 biomedicines-10-02785-t003:** Results of the cognitive test (mean and SD).

Variable/Score	Minimum	Maximum	Mean ± SD
Reasoning	227.0	717.0	474.8 ± 99.2
Memory	153.0	647.0	385.5 ± 115.8
Attention	196.0	745.0	554.4 ± 112.2
Coordination	10.0	649.0	289.7 ± 132.6
Perception	249.0	672.0	473.2 ± 89.1
Total Cognitive function	221.0	614.0	440.7 ± 83.2

**Table 4 biomedicines-10-02785-t004:** The relative risk for cognitive functions in the first trimester.

	Reasoning RR (95% CI)	Memory RR (95% CI)	Attention RR (95% CI)	Coordination RR (95% CI)	Perception RR (95% CI)	Total Cognitive Function RR (95% CI)
Age (years)	1.269 (0.397 4.058)	1.026 (0.661 1.592)	3.750 (0.474 29.637)	1.005 (0.783 1.292)	1.625 (0.644 4098)	1.161 (0.523 2.577)
Onset T1DM after ten years	0.670 (0.213 2.102)	0.755 (0.492 1.158)	3.360 (0.426 26.517)	0.824 (0.654 1.040)	1.456 (0.579 3.660)	1.040 (0.470 2.300)
Duration T1DM > 8 years	2.938 (0.734 11.759)	1.704 (0.933 3.110)	2.509 (0.320 19.692)	1.027 (0.748 1.410)	1.087 (0.438 2.699)	1.309 (0.537 3.192)
BMI (kg/m^2^)	1.273 (0.382 4.238)	2.114 (1.138 3.927)	0.883 (0.184 4.234)	1.461 (1.031 2.071)	1.405 (0.622 3.177)	2.208 (1.116 4.370)
HbA1c in1st trimester	0.597 (0.168 2.119)	1.156 (0.761 1.756)	0.473 (0.092 2.422)	1.017 (0.824 1.256)	0.394 (0.155 1.004)	0.774 (0.366 1.638)
Hypothyroidism	2.750 (0.786 9.616)	1.304 (0.830 2.048)	1.333 (0.238 7.481)	0.889 (0.350 2.260)	0.889 (0.350 1.260)	3.111 (1.140 8.491)
C-peptide concentration in 1st trimester (pmol/L)	0.222 (0.055 0.895)	0.450 (0.210 0.958)	0.500 (0.059 4.232)	0.696 (0.487 0.993)	0.462 (0.145 1.466)	0.297 (0.097 0.912)
CAN	4.167 (1.238 14.022)	1.647 (1.055 2.572)	1.750 (0.421 5.273)	1.433 (1.075 1.911)	1.021 (0.426 2.447)	2.250 (1.000 5.062)
Education	0.375 (0.111 1.271)	1.320 (0.841 2.073)	1.073 (0.258 4.462)	1.260 (0.979 1.621)	0.715 (0.310 1.649)	1.380 (0.613 3.107)
CGM > 7.8 mmol/L	0.833 (0.209 3.323)	1.000 (0.578 1.730)	1.667 (0.303 9.157)	0.972 (0.730 1.295)	0.750 (0.233 2.413)	1.875 (0.757 4642)
CGM < 3.9 mmol/L	1.460 0.342 6.226)	1.696 (0.794 3.623)	1.429 (0.466 4.376)	1.097 (0.783 1.536)	1.087 (0.344 3.437)	0.938 (0.346 2.539)

## Data Availability

Data supporting reported results can be found in: https://figshare.com/articles/dataset/DATA_pdf/20677170, accessed on 27 August 2022.

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
