# Peer review of "Positive Association between Preserved C-Peptide and Cognitive Function in Pregnant Women with Type-1 Diabetes"

_biomedicines, 2022, doi:10.3390/biomedicines10112785_

Round 1

Author Response

This paper refers to the cognitive function assessment in pregnant women with type 1 diabetes and its relationship with other variables, such as preserved C-peptide, obesity, hypothyroidism, and long-term diabetic complications including cardiovascular autonomic neuropathy. The topic of the study is of great importance because it is well-documented that both diabetes and pregnancy may impair cognitive functions. The manuscript adds a significant contribution to the field, there are however some discrepancies and inaccuracies that are listed below:

  1. The abstract should be carefully revised because it is confusing, especially in lines 20-24. The reader doesn’t know what exactly the authors wanted to say. Please clarify.
  2. Relative risks for low scores for general cognitive function were associated with increased BMI >25 kg/m2 2.208 (95% CI 1.116-4.370), HbA1c >6.5% RR 0.774 (95% CI 0.366-1.638), subclinical hypothyroidism RR 3.111 (95% CI 1.140-8.491), and impaired cardiovascular autonomic neuropathy RR 2.250 (95% CI 1.000-5.062). Pregnant women with a lower score for general cognitive function had higher BMI and leptin levels.
  3. The introduction from line 50 is a bit messy. I understand that the authors are trying to refer to every studied parameter, but it should be connected, one thing should be related to another, because in that form it is confusing for the reader
  4. Patients who have complications of diabetes, such as proliferative retinopathy, nephropathy, neuropathy, hypertension, and cardiovascular autonomic neuropathy, have poorer cognitive function compared to those without complications. The development of the complications mentioned above and changes in the brain depends on the duration of diabetes and the quality of metabolic control. Based on previous research, it can be concluded that cognitive impairment is one of the complications of type 1 diabetes.
  5. Figure 1 should be corrected – the caption is incomplete, which one is a and b? the last line on both parts of the figure is not signed – what parameter is it?
  6. It was corrected as suggested.
  7. My main concern regarding the study is the timing of cognitive function testing – 10 weeks gestation. I wonder if this is the best time for the assessment – there are available in the literature studies concerning cognitive function in pregnancy and some of them showed it is affected in the second and third trimesters and in the postpartum period and the impact of early pregnancy on cognition might be very small and debatable. Here are the references that are not included in the manuscript, but might be important: Farrar D, Tuffnell D, Neill J, et al. Assessment of cognitive function across pregnancy using CANTAB: a longitudinal study. Brain Cogn. 2014;84:76–84., Yee LM, Kamel LA, Quader Z, et al. Characterizing Literacy and Cognitive Function during Pregnancy and Postpartum. Am J Perinatol. 2017;34:927–934. Grattan DR, Ladyman SR. Neurophysiological and cognitive changes in pregnancy. Handb Clin Neurol. 2020;171:25–55.
  8. Cognitive function is significantly worse in pregnant women than in control women, which some authors associate with increased hormones, progesterone, estrogen, human placental lactogen (HPL), cortisol, prolactin (PRL), and others. Our participants with T1DM had their cognitive function tested in the first trimester of pregnancy due to the more frequent occurrence of hypoglycemia than in the second and third trimesters. However, the study did not show an association between lower glucose values measured by CGM and a higher frequency of lower scores for cognitive function.
  9. The lack of a control group is listed as the limitation of the study and I fully agree with it. Moreover, in my opinion, authors should not write that they didn’t have the opportunity to compare this study with other research results – they should refer to other studies concerning cognition in pregnancy, for example, those listed by me in point 4. The most important question is if the cognition impairment is related to diabetes, pregnancy, or both of them, and the presented study does not answer this question. The authors should discuss it in detail at the end of their discussion.
  10. The cognition impairment in T1DM is related mainly to diabetes and partly to pregnancy.
  11. For lines 254-259 please provide appropriate references

It was done.

  1. Lines 263-265 rephrase the sentence: “overweight and obese pregnant women are a risk…”
  2. This sentence was removed because it was redundant.

The authors also found a negative correlation between cognitive function score and leptin concentrations – a short explanation/hypothesis of a possible origin of that finding should be provided to improve the discussion.

  1. We explained in more detail as suggested. Leptin is an adipostatic hormone that is released into the circulation depending on the amount of adipose tissue It participates in the regulation of body weight through the suppression of appetite and stimulation of energy consumption. Leptin is strongly correlated with maternal weight and BMI. It is a biomarker of maternal and fetal obesity.

  1. Please pay more attention to the abbreviations – not all of them are explained, e.g. CAN

Thank you, we corrected all abbreviations.

Reviewer 2 Report

Thank you for allowing me to review your manuscript. It is a novel and interesting study. However, I have some suggestions on improvements before it can be published:

Abstract: Lines 23-24. "In women with C-peptide preservation...." this sentence is incomplete.

Introduction: Lines 69-71. Is this your study aim/objective? This needs to be clearly stated.

Materials and methods:

Line 81. How was diabetes diagnosed?

Lines 93-94. Why were women with these conditions excluded from the study? I would suggest that you provide a reason and cite the literature.

Lines 164-165. Was there a literature search for the meta-analysis? What information sources were searched? Limitations on search? Authors who abstracted data? PRISMA guidelines followed? Inclusion and  exclusion criteria for meta-analysis? Fixed effects or random effects model used for meta-analysis? There is no mention of meta-analysis in the abstract or title of the manuscript. 

Results: 

Table 1. Indicate that X is "mean" and SD is "standard deviation". Acronyms e.g., CGM and CAN, and TSH should be written out as a footnote under the table. 

Table 4. Were all exposure variables included in the same statistical model? Or were they analyzed in separate models?

Figure 1. The authors should indicate that these are the results from the meta-analysis. Also, the Forest plots do not have explanations or labels as to what the numbers 0.1 1, and 10 stand for?  Do they show the mean difference? That should be clearly labeled. Heterogeneity? Chi-square test for heterogeneity? I-squared value? Characteristics of studies used in meta-analysis, i.e., sample size, demographics should be included in a separate table. 

Discussion:

Lines 287-292. Limitations of the meta-analysis should be mentioned. 

Conclusion:

Lines 294-296. Based on the findings of this study, how will this study improve cognitive function among pregnant women with diabetes? What gaps were addressed? 

Author Response

Thank you for allowing me to review your manuscript. It is a novel and interesting study. However, I have some suggestions for improvements before it can be published:

  1. Abstract: Lines 23-24. "In women with C-peptide preservation...." this sentence is incomplete.
  2. Several sentences have been changed in the abstract, as follows: Relative risks for low scores for general cognitive function were associated with increased BMI >25 kg/m2 2.208 (95% CI 1.116-4.370), HbA1c >6.5% RR 0.774 (95% CI 0.366-1.638), subclinical hypothyroidism RR 3.111 (95% CI 1.140-8.491), and impaired cardiovascular autonomic neuropathy RR 2.250 (95% CI 1.000-5.062). Pregnant women with a lower score for general cognitive function had higher BMI and leptin levels.
  3. Lines 69-71. Is this your study aim/objective? This needs to be clearly stated.
  4. The study aimed to investigate the impact of age, duration of diabetes, BMI, subclinical hypothyroidism, cardiovascular autonomic neuropathy, hypo-/hyperglycemia, and C-peptide preservation on cognitive performance.

Materials and methods:

  1. Line 81. How was diabetes diagnosed?
  2. All women with T1DM have been diagnosed before pregnancy. The diagnosis was confirmed if the autoantibodies GADA and IAA were positive.
  3. Lines 93-94. Why were women with these conditions excluded from the study? I would suggest that you provide a reason and cite the literature.
  4. This study is part of the scientific project PRE-HYPO as a part of the Croatian Science Foundation. According to the project, the exclusion criteria were proliferative retinopathy, nephropathy, and chronic hypertension in type 1 diabetic pregnant women.
  5. Lines 164-165. Was there a literature search for the meta-analysis? What information sources were searched? Limitations on search? Authors who abstracted data? PRISMA guidelines followed? Inclusion and exclusion criteria for meta-analysis? Fixed effects or random effects model used for meta-analysis? There is no mention of meta-analysis in the abstract or title of the manuscript. 
  6. No, the authors did not perform a meta-analysis of other authors' results, they used the Comprehensive Meta-Analysis program for our own results only so the PRISMA guidelines are not applicable to this study.

Results: 

  1. Table 1. Indicate that X is "mean" and SD is "standard deviation". Acronyms e.g., CGM and CAN, and TSH should be written out as a footnote under the table. 
  2. It was corrected as suggested.
  3. Table 4. Were all exposure variables included in the same statistical model? Or were they analyzed in separate models?
  4. Most of the statistical analysis was done using the SPSS program, for RR, forest plots, and heterogeneity using the statistical program Comprehensive meta-analyses.
  5. Figure 1. The authors should indicate that these are the results from the meta-analysis. Also, the Forest plots do not have explanations or labels as to what the numbers 0.1 1, and 10 stand for.  Do they show the mean difference? That should be clearly labeled. Heterogeneity? Chi-square test for heterogeneity? I-squared value? Characteristics of studies used in the meta-analysis, i.e., sample size, and demographics should be included in a separate table.
  6. Thank you for this valuable suggestion. It was done as recommended. 

Discussion:

  1. Lines 287-292. Limitations of the meta-analysis should be mentioned. 
  2. In this study, we used meta-analysis, which has its limitations. For memory and overall cognitive function, heterogeneity was I2 48.6% and 40.7%, which represents moderate heterogeneity.

Conclusion:

  1. Lines 294-296. Based on the findings of this study, how will this study improve cognitive function among pregnant women with diabetes? What gaps were addressed? 
  2. Good metabolic control, normal BMI, normal thyroid function, and preserved C-peptide are of essential importance for preserved cognitive function in pregnant women with T1DM.

Round 2

Reviewer 1 Report

the authors improved the manuscript according to reviewer's suggestion 

Reviewer 2 Report

Thank you very much for revising the manuscript according to my suggestions. I have no further comments.